# The Stochastic Nature Exhibited by Proteins inside the Cell Membrane during Cell-to-Cell Communication

**DOI:** 10.3390/biology12081102

**Published:** 2023-08-08

**Authors:** Nick Malope, Ebrahim Momoniat, Rhameez Sheldon Herbst

**Affiliations:** Department of Pure and Applied Mathematics, University of Johannesburg, Johannesburg 2006, South Africa

**Keywords:** cell-to-cell communication, messenger proteins, *Itô* Lemma, stochastic diffusion equation, cell membrane, Brownian motion

## Abstract

**Simple Summary:**

This paper presents a brief mathematical study in which the communication between cells is modelled. Understanding cell communication mathematically is useful across many medical fields from drug delivery to understanding how the body responds to infection. In this paper, the random motions of proteins are explored using stochastic differential equations. The study demonstrates the Brownian motion exhibited by proteins during cell communication.

**Abstract:**

The movement of proteins through the cell membrane is essential for cell-to-cell communication, which is a process that allows the body’s immune system to identify any foreign cells, such as cells from another organism and pathogens; this movement is also essential for protein-to-protein interactions and protein-to-membrane interactions which play a significant role in drug discovery. This paper presents the stochastic nature exhibited by proteins during cell-to-cell communication. We study the movement of proteins through the cell membrane under the influence of an external force *F* and drag force with drag coefficient γ. We derive the stochastic diffusion equation, which governs the motion of the proteins; we start by describing the random motion exhibited by the proteins in terms of probability using a one-dimensional lattice model; this occurs when proteins move inside the cell membrane and bind with other proteins inside the cell membrane. We then introduce an external force and a drag coefficient into a Brownian motion description of the movement of proteins when they move outside the cell membrane and bind with proteins from other cells; this phenomenon occurs during cell communication when one cell releases messenger proteins to relay information to other cells. This, in turn, allows us to obtain the stochastic diffusion equation by applying $It\hat{o}$’s Lemma.

## 1. Introduction

This paper aims to offer a mathematical description of cell-to-cell communication and supplements the biological understanding of cell-to-cell communication by introducing a model that describes this phenomenon. Cell-to-cell recognition is “the initial event in cell-to-cell communication that evokes a defined biochemical, physiological or morphological response” [1]. In order to maintain stable cellular homeostasis, cells need to exchange information with each other constantly and their surrounding environment; this function is facilitated by proteins in the cell membranes, which critically relay information within cells and the cell environment [2]. Cell-to-cell communication is the primary function for cells to recognize each other; cells such as natural killer cells constantly undergo cell-to-cell communication to identify mutated cells, such as tumor cells and cells infected with viruses, and terminate these cells [3]. Scientists need to understand the behavior of cells during cell-to-cell communication, precisely, how cells move and relay information to each other and their surrounding environment. In the past years, there have been significant advancements in the study of cell-to-cell communication; the brilliant work by Martin Benoit and Hermann E. Gaub [4] in quantifying adhesion forces experienced by cells during cell-to-cell communication using atomic force microscopy highlighted the need for quantitative description of cell-to-cell communication. The failure of cell-to-cell communication initiates and promotes the spread of diseases in the body, and the weakening of cell-to-cell adhesion has a high correlation with the survival and growth of cancer tumor cells [5]. The binding failure of polymorphonuclear neutrophils from non-adhesion of Leukocytes promotes ischemic renal acute failure [6]; the failure of cell-to-cell communication affects dozens of people worldwide and gives rise to an unlimited number of diseases and disrupts critical bodily functions at a cellular level.

## 2. Materials and Methods

To study how cell-to-cell communication occurs, we must first study the motion of proteins inside the cell membrane; this motion of proteins gives rise to many bodily functions, from hormone production to hemostasis and disease prevention [7]. Proteins are found in the cell membrane, which separates the interior of the cells from the outside environment. The cell membrane protects cells and provides a fixed environment within the cell; some of the main functions of the cell membrane are to transport nutrients into the cell, transport toxic substances out of the cells, and allow proteins to pass through the cell membrane [8]. We shall study the motion of proteins inside the cell membrane, followed by the study of proteins when they diffuse through the cell membrane to the exterior of the cell so that they can bind with other proteins from other cells during cell communication. Fick’s law of diffusion describes the macroscopic diffusion of particles from a region of higher concentration to a level of lower concentration, with a diffusion rate being the concentration gradient [9]; the macroscopic view of Fick’s law does not take into account the microscopic view of diffusion, especially when considering diffusion within a cell membrane. At a microscopic level, the diffusion of proteins in the cell membrane is highly stochastic [9].

We consider a protein inside a cell membrane that can move at discrete times between neighboring sites on a one-dimensional lattice with unit spacing [9] (see Figure 1); At each step, the protein moves a unit distance to the right with probability *p* or to the left with probability q=1−p. Let $P(x_{i},n)$ denote the probability that the protein is at location $x_{i}$ at the *n* th time step. The evolution of the probability distribution is described by the discrete-time equation given by
(1)$$P\left(x_{i},n\right)=pP\left(x_{i-1},n-1\right)+qP\left(x_{i+1},n-1\right)\;x_{i},n\in Z,n\geq 1$$

We introduce the characteristic equation by taking the Discrete Fourier Transform on both sides of Equation (Equation 1) for fixed *n*.
(2)$$∴{\hat{P}}_{n}\left(k\right)=\left(pe^{-jk}+qe^{jk}\right){\hat{P}}_{n-1}\left(k\right)$$

To solve Equation (Equation 2), we need to know what the value ${\hat{P}}_{0}\left(k\right)$ is; let us assume, without loss of generality that ${\hat{P}}_{0}\left(k\right)=1$. Thus, then Equation (Equation 2) becomes
(3)$$∴{\hat{P}}_{n}\left(k\right)={\left(pe^{-jk}+qe^{jk}\right)}^{n}$$

Taking the Inverse Fourier Transform of Equation (Equation 3),
(4)$$P\left(x_{i},n\right)=\frac{1}{2\pi }\int_{-\pi }^{\pi }e^{-jkx_{i}}{\left(pe^{-jk}+qe^{jk}\right)}^{n}dk$$
from the binomial theorem, we can write Equation (Equation 4) as
(5)$$P\left(x_{i},n\right)=\sum_{l=0}^{n}\frac{e^{\left(2l-n-x_{i}\right)}n!}{(n-l)!l!}p^{n-l}q^{l}$$

For infinitesimal step length $\delta x=\frac{x_{i}}{N_{x}}$ and time length $\delta t=\frac{t}{N_{t}}$, where $N_{x}$ denotes the number of points along the one-dimensional lattice, and $N_{t}$ is the number of time points, we define the probability per unit length as the probability density function given by
(6)$$\rho \left(x,t\right)=\frac{P(x_{i},n)}{\delta x}$$We substitute Equation (Equation 6) into Equation (Equation 1),
(7)⇒ρ(x,t)=pρ(x−δx,n−δt)+qρ(x+δx,n−δt)
using a Taylor Series expansion with a first order of δt and second-order δx [10],
(8)$$∴\frac{\partial \rho (x,t)}{\partial t}\delta t=\left(q-p\right)\delta x\frac{\partial \rho (x,t)}{\partial t}+\frac{\delta x^{2}}{2}\frac{\partial^{2}\rho \left(x,t\right)}{\partial x^{2}}$$We divide Equation (Equation 8) by δt,
(9)$$\frac{\partial \rho (x,t)}{\partial t}=\left(q-p\right)\frac{\delta x}{\delta t}\frac{\partial \rho (x,t)}{\partial t}+\frac{\delta x^{2}}{2\delta t}\frac{\partial^{2}\rho \left(x,t\right)}{\partial x^{2}}$$Due to the fact that proteins move extremely fast [11], the step length δx and the time δt it takes for a protein to move from one point to another become significantly small; thus, we can take limδt,δx→0,
(10)$$\frac{\partial \rho (x,t)}{\partial t}=\lim_{\delta t,\delta x\to 0}\left(q-p\right)\frac{\delta x}{\delta t}\frac{\partial \rho (x,t)}{\partial t}+\lim_{\delta t,\delta x\to 0}\frac{\delta x^{2}}{2\delta t}\frac{\partial^{2}\rho \left(x,t\right)}{\partial x^{2}}$$
we denote $\psi =\lim_{\delta t,\delta x\to 0}\left(q-p\right)\frac{\delta x}{\delta t}$ and $D=\lim_{\delta t,\delta x\to 0}\frac{\delta x^{2}}{2\delta t}$.

Consider the case where the drift velocity ψ=0, and initial condition ρ(x,0)=f(x),
(11)$$\frac{\partial \rho (x,t)}{\partial t}=D\frac{\partial^{2}\rho \left(x,t\right)}{\partial x^{2}}$$
applying the Fourier transform,
(12)$${\hat{\rho (k,t)}}_{t}=D{\left(ik\right)}^{2}\hat{\rho (k,t)}$$
(13)$$∴{\hat{\rho (k,t)}}_{t}+Dk^{2}\hat{\rho (k,t)}=0$$

Multiplying by the integrating factor, Equation (Equation 13) becomes
(14)$$\frac{\partial }{\partial t}\left(e^{Dk^{2}t}\hat{\rho (k,t)}\right)=0$$
integration with respect to *t* yields
(15)$$\hat{\rho (k,t)}=e^{-Dk^{2}t}f\left(k\right)$$

The Inverse Fourier Transform yields
(16)$$\rho \left(x,t\right)=\frac{1}{\sqrt{4\pi Dt}}\int_{-\infty }^{\infty }e^{-\frac{{\left(x-z\right)}^{2}}{4Dt}}f\left(z\right)dz$$

This is the probability density function describing the likelihood of the protein being located at point *x*. Consider the random motion of a protein inside a cell membrane, subjected to external forces *F*, with drag coefficient γ. Let $X_{t}$ be the position of the protein at time *t* inside the cell membrane; we define the infinitesimal change in the position of the protein as it moves through the cell membrane in terms of a random variable that is normally distributed by the process:(17)$$dX_{t}=\frac{F}{\gamma }X_{t}dt+\sqrt{D}X_{t}\sqrt{dt}dZ$$
where we have used the fact that $dW=\sqrt{dt}dZ$, dZ∼N(0,1), $dW_{t}∼N\left(0,dt\right)$ and $dX_{t}∼N\left(0,Ddt\right)$ are random variables that are normally distributed with mean=0 and variances = dt [12]. We introduced the diffusivity *D*, which is the rate of diffusion that controls the volatility of the process, a measure of the rate at which proteins diffuse into and out of the cell membrane; we also introduced the drift term $\frac{F}{\gamma }$.

We set the initial position of the protein at time t=0, $X_{0}$, to zero, consider the time interval t∈[1,T], and we divide the time *T* into small *n* equal interval $dt=\frac{T}{n}$, since the position of the protein changes by the amount given by $dX_{i}=\frac{F}{\gamma }X_{i}dt+\sqrt{Ddt}X_{i}dZ_{i}$, where dZ∼N(0,1) over the *i*th time interval; the final position $X_{T}$ is given by,
(18)$$X_{T}=\frac{F}{\gamma }\sum_{i=1}^{n}X_{i}dt+\sqrt{Ddt}\sum_{i=1}^{n}dZ_{i}$$

We now have a stochastic description of the motion of a protein in a cell membrane. The behavior and motion of the proteins determine the success of cell-to-cell communication; the exact solution of Equation (Equation 17) is found by applying Ito’s formula and solving for $X_{t}$. The analytical solution is given by,
(19)$$X_{t}=x_{0}\cdot e^{\left(\frac{F}{\gamma }-\frac{1}{2}D\right)t-\sqrt{D}W_{t}}$$
where $X_{0}=x_{0}$; let Θ(X,t) be a function of the stochastic function *X* and time *t*.

From Ito’s Lemma, we can determine the process followed by the function Θ(X,t) by applying Taylor expansion up to second order in variables *X* and *t* [12].
(20)$$\Theta \left(X+dX,t+dt\right)=\Theta +\frac{\partial \Theta }{\partial t}dt+\frac{\partial^{2}\Theta }{2\partial t^{2}}dt^{2}+\frac{\partial \Theta }{\partial X}dX+\frac{\partial^{2}\Theta }{2\partial X^{2}}dX^{2}+\frac{\partial \Theta }{\partial X\partial t}dXdt$$
(21)$$\frac{d\Theta }{dt}=\frac{\partial \Theta }{\partial t}+\frac{\partial^{2}\Theta }{2\partial t^{2}}dt+\frac{\partial \Theta }{\partial X}\frac{dX}{dt}+\frac{\partial^{2}\Theta }{2\partial X^{2}}\frac{dX^{2}}{dt}+\frac{\partial \Theta }{\partial X\partial t}dX$$
where dΘ=Θ(X+dX,t+dt)−Θ(X,t).

We substitute Equation (Equation 17) into Equation (Equation 21), and take the limit dt→0, in Equation (Equation 21). The limit dt→0 is computed because the time it takes for a protein to move from one location to another becomes significantly small as proteins move at high speeds; we obtain the below equation:(22)$$\lim_{dt\to 0}\frac{d\Theta }{dt}=\frac{\partial \Theta }{\partial t}+\left(\frac{F}{\gamma }X+\sqrt{\frac{D}{dt}}XdZ\right)\frac{\partial \Theta }{\partial X}+DX^{2}dZ^{2}\frac{\partial^{2}\Theta }{2\partial X^{2}}$$

This is the change in value for Θ(X,t) over the infinitesimal time interval dt. The stochastic behavior of proteins allows us to study the interaction of one protein with another protein; this mainly serves as the receiving end of chemical reactions, as they float freely inside and outside the cell membranes during cell communication, exhibiting Brownian motion behavior.

We transform Equation (Equation 22) into a diffusion equation by taking the limit dt→∞ [12], where $dZ^{2}$ is equal to 1.
(23)$$∴\frac{\partial \Theta }{\partial t}+\frac{F}{\gamma }X\frac{\partial \Theta }{\partial X}+\frac{D}{2}X^{2}\frac{\partial^{2}\Theta }{\partial X^{2}}=0$$

As suggested by Paul Wolfgang [13], in order to make Equation (Equation 22) dimensionless, we make use of the following change in variables, $X=Ke^{x}$, $\tau =T\frac{\sigma^{2}}{2}-t\frac{\sigma^{2}}{2}$, Θ=Kθ(x,τ), where *K* is a scaling factor, and $\sigma =\sqrt{D}$ is the volatility of the process, we obtain the below equation:(24)$$∴\frac{-K\sigma^{2}}{2}\frac{\partial \theta }{\partial \tau }+\frac{FK}{\gamma }\frac{\partial \theta }{\partial x}+\frac{DK}{2}\left(\frac{1}{X^{2}}\frac{\partial \theta^{2}}{\partial x^{2}}-\frac{1}{X^{2}}\frac{\partial \theta }{\partial x}\right)=0$$

Inside the cell membrane, we define the diffusivity rate to be proportional to the external forces acting on the protein *F* and the drag coefficient γ by a simple relationship given by
(25)$$D=2{\left(\frac{F}{\gamma }\right)}^{2}$$

However, Dan V. Nicolau Jr. et al. [14] suggested that the diffusivity inside a cell membrane, is spatially dependent and is given by
(26)$$D\approx \frac{e^{2x}\Gamma \left(1+\alpha \right)}{2}$$
where Γ(x) is the Gamma function. For maximum diffusivity, we set α=1,
(27)$$∴D\approx \frac{e^{2x}}{2}$$
(28)$$\Rightarrow F\approx \frac{\gamma e^{x}}{2}$$
(29)$$∴\frac{\partial \theta }{\partial \tau }=\left(\frac{e^{2x}}{2\sigma^{2}}\right)\frac{\partial^{2}\theta }{\partial x^{2}}+\frac{e^{x}}{\sigma^{2}}\left(1-\frac{e^{x}}{2}\right)\frac{\partial \theta }{\partial x}$$

This is the diffusion advection equation subjected to the Dirichlet boundary conditions θ(0,τ)=θ(1,τ)=0 and initial condition $\theta \left(x,0\right)=e^{2x}\left(x-x^{2}\right)$.

## 3. Results and Discussion

In this section, we present and discuss the numerical results of Equations (Equation 17) and (Equation 29); the results are computed in R using the *diffeqr*, *ReacTran* and *plotly* libraries. Figure 2 shows the numerical simulations of Equation (Equation 17) with varying external forces *F*; the force varies from $F=1\times 10^{-6}\;\mu$N to F=1μN with all other factors kept constant [15]. These external forces regulate the cellular membrane so that it adjusts to the changing cellular environment by controlling the cell shape. The numerical simulation shows that the random motion exhibits an exponential motion; this motion is highlighted in the analytical solution in Equation (Equation 19). The movement of proteins is evident from the fluid mosaic model; this motion is important in cell communication; the external forces and the drag force facilitate this motion, and the physical forces at the plasma membrane trigger nanoscale membrane deformations that are then translated into chemical signals [15]. The relationship of the external force with the drag coefficient is highlighted in Equation (Equation 28). Equation (Equation 29) is solved numerically subjected to the boundary and initial conditions stated in Equation (Equation 29); the simulations are carried out for different values of σ, and the solutions are presented in line plots of the diffusion advection equation. We start by varying the value of σ from σ=5 to σ=20; the diffusivity and advective terms in Equation (Equation 29) are spatially dependent exponential functions implying that diffusion grows quicker with varying spatial values. The function θ(X,t), which is a function of the stochastic process *X* and time *t*, is a function that describes the diffusion and advection of proteins into and out of the cell membrane as supported by the fluid mosaic model. This movement is essential in cell communication; it results in successful cell binding and also allows cells to recognize foreign cells such as cancerous cells.

Consider Equation (Equation 17); this equation is solved numerically as depicted in Figure 2 using the Euler–Maruyama method on the interval [a,b], where a=0,b=1 by assigning the grid points $a=t_{0}<t_{1}<t_{2}<\ldots <t_{n}=b$ with estimated x-values $X_{0}<X_{1}<X_{2}<\ldots <X_{n}$, where $X_{i}=X\left(t_{i}\right)$, we solve the initial value problem of Equation (Equation 17) subjected to the initial condition $X_{0}=x_{0}$; we represent this equation as follows:(30)$$X_{i+1}=X_{i}+\frac{F}{\gamma }X_{i}\Delta t_{i}+\sqrt{D}X_{i}\Delta W_{i}$$
where $\Delta t_{i}=t_{i+1}-t_{i}$, $\Delta W_{i}=W_{i+1}-W_{i}$; recall that $dW=dZ\sqrt{dt}$ where Z∼N(0,1), $∴\Delta W_{i}=\Delta Z_{i}\sqrt{\Delta t_{i}}$, the set of the solutions $\left\{X_{0},X_{1},X_{2},\;\ldots ,\;X_{n}\right\}$ is the approximate solution to the stochastic differential equation X(t) which depends on the random values dZi. The exact solution in Equation (Equation 17) and the numerically simulated Equation (Equation 29) are exponential; recall that each computed point $X_{i}$ is only an approximation to the stochastic differential equation X(t). Each of these values results in a random variable *T*, such that the difference between the exact solution and the approximated solution converges strongly if the expected value of the difference converges to zero, i.e., ϵ=Y(T)−X(T), if $\lim_{t\to 0}E\left(\epsilon \right)=E\{|Y\left(T\right)-X\left(T\right)|\}$; then, the solution converges strongly, where Y(t) is the exact solution of the stochastic differential equation [16].

Consider the differential equation in Equation (Equation 29); this is a convection–diffusion equation where the convection and diffusivity terms are spatially dependent; we simulate this equation for varying volatility σ, and show the results as line plots in Figure 3, where the dark bold line represents the initial condition. Consider the limit σ→0; the diffusivity and convection term becomes significantly prominent in the equation; the solution rapidly takes on the values imposed by the boundary condition closer to the upstream value *x* = 1. For the limit σ→∞, the solution becomes more consistent with the initial conditions imposed.

The function Θ(X,t) is a function of the stochastic function *X* and time *t*; it is a function that describes the diffusion of proteins as a function of the protein’s stochastic motion for successful protein binding during cell communication. The movement of proteins allows the proteins to recognize each other and to also relay information with each other and recognize foreign cells; proteins travel through diffusion and convection. The concept of diffusion is not only limited to proteins, but it also allows other molecules to find each other and bind successfully, such as substrates locating enzymes and DNA binding to proteins; thus, it is clear that diffusion is essential in the overall functioning and well-being of cells. It is worth mentioning that cells are densely packed with proteins, ribosomes, and other biological molecules, such that the distance between two biological molecules is significantly minimal; due to thermal motion, proteins move extremely fast, reaching speeds of about 400 km/h [11]. This means that delays in cell signaling are significantly minor; the proteins constantly collide with other proteins, thus exhibiting random motion. Because of this high-speed random motion and frequent molecular collision, it is complicated to simulate the reality of the motion exhibited by proteins inside the cell membrane.

## 4. Conclusions

We presented a stochastic model of protein movement inside the cell membrane under the influence of an external force and drag force; we showed that the motion of proteins inside cell membranes is Brownian in nature and simulated this motion; we also simulated the stochastic diffusion motion of proteins when they diffuse through cell membranes. During cell-to-cell communication, cells transmit signals differently to interact with their environment and other cells. Cells may communicate with nearby cells by direct signaling or with cells further away using endocrine signaling, where signaling proteins are transmitted through the bloodstream. The mathematical study of cell-to-cell communication and signaling is critical in understanding how cells communicate with each other from a mathematical point of view in medicine. This allows us to understand the possible causes of cell-to-cell recognition and signaling failure, enabling us to understand how the body’s immune system does not pick up some pathogenic cells or why they may be detected later by the body’s cells. The model presented in this paper focuses on the behavior of proteins that are transmitted during cell communication; the model can be expanded further to study protein-to-protein interactions and specific cell-to-cell communication methods such as endocrine signaling methods by incorporating fluid mechanics principles for proteins that are transmitted via the bloodstream and to study specific interactions such as ligand–receptor interactions. Furthermore, we can now ask and answer fundamental questions such as how to improve cell communication and cell recognition and increase foreign cell detection early. The ability of cells to detect foreign cells, such as cancerous cells, is a crucial function of cell-to-cell communication as it allows the body to eliminate pathogenic cells at an early stage. The constant movement of proteins inside the cell membrane is another vital mechanism that can be improved; this is because when proteins inside the cell membrane move, they have an increased chance of binding to each other and relaying cellular information to other cells; thus, we need to find a medically sane method of improving and increasing the motion of proteins inside the cell membrane and determine the implications this would have in the body at a cellular level.

## Figures and Tables

**Figure 1 biology-12-01102-f001:**
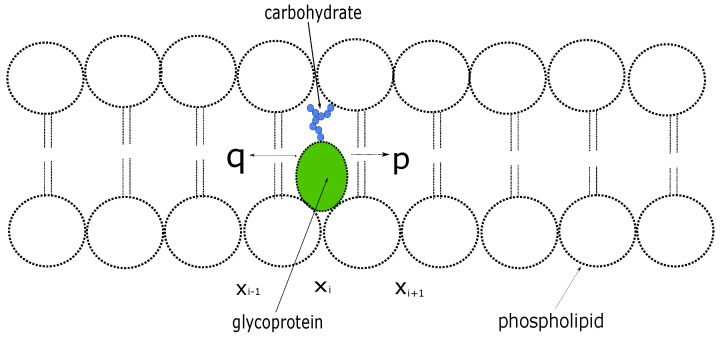
Glycoprotein inside a cell membrane.

**Figure 2 biology-12-01102-f002:**
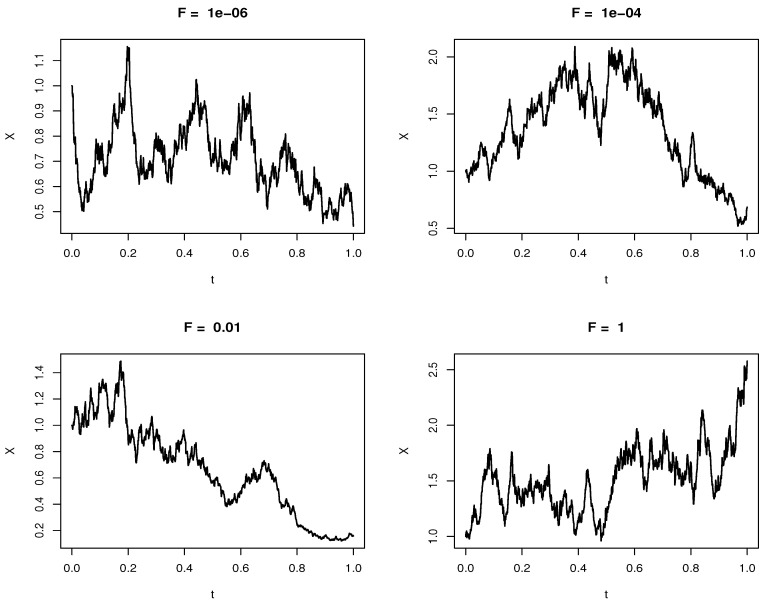
Simulation of the Brownian motion of exhibited by glycoproteins in Equation (Equation 17) for varying external forces *F*.

**Figure 3 biology-12-01102-f003:**
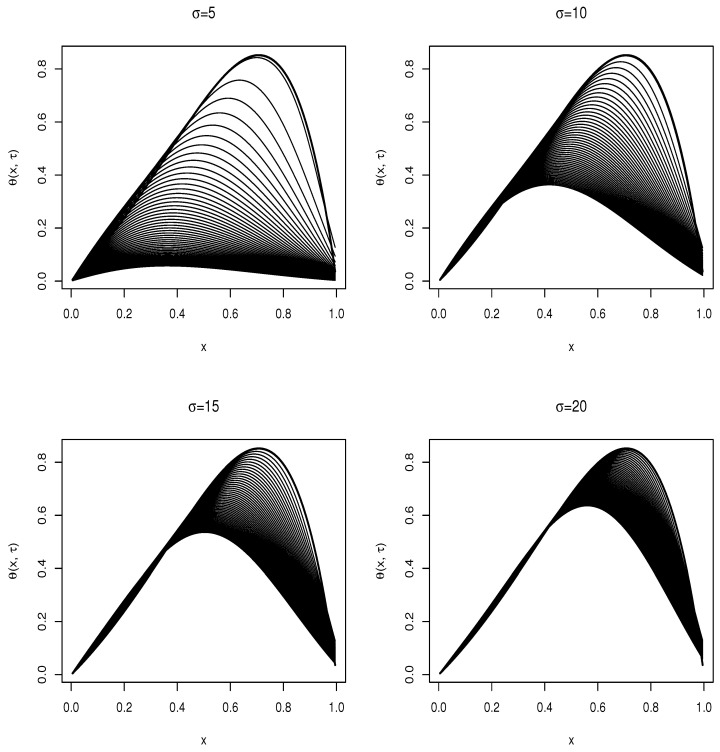
Line plots of the solution of the diffusion advection Equation (Equation 29) for varying volatility σ.

## Data Availability

No public involvement in any aspect of this research.

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
