# Peer review of "The Stochastic Nature Exhibited by Proteins inside the Cell Membrane during Cell-to-Cell Communication"

_biology, 2023, doi:10.3390/biology12081102_

Round 1

Reviewer 1 Report

I read with great interest the paper by Malope et al. The authors present a mathematical approach to the diffusion of a protein molecule inside a cell membrane. The mathematical model is stochastic rather than deterministic in nature, which is appropriate for describing the motion of one protein molecule.  

Major comments: 

1.     The authors talk about cell-cell communication and that the model is relevant. However, cell-cell communication is very general term, which encompasses several processes, which are not captured by this model. For instance, ligand-receptor interactions are not described by this approach. The point is that the description should be more detailed, and the authors should explain which specific physiological processes this model applies to.

2.     Following up on the previous point, protein interactions are possible. Is there a way to expand the model to take into account such interactions? The authors are not asked to extend the model, but include in the discussion how this may happen.

3.     The ‘Conclusions’ section also contains many general forward-looking statements instead of the main take-aways of the study.

4.     Describing in the results the physiological meaning or relevance of the different parameters (e.g., the constant K, the sigma etc.) would be helpful for better interpretation of the graphs presented in figures 3 and 4. 

5.     The code(s) should be made available through a public data repository, per the journal’s guidelines.

6.     The steps involving equations (20)-(23) in light of dt approaching zero should be described with more clarity.  

The manuscript will benefit significantly for vetting for language. 

Author Response

The authors would like to thank the reviewer for their helpful suggestions. 

Please see the list of comments addressed in the report. 

Many thanks

Reviewer 2 Report

The authors have presented a simple 1D drift-diffusion model to describe the movement of a protein molecule inside a cell membrane. The model is not novel, and this manuscript has many significant concerns (see the points below). Hence, I cannot recommend publication in the present form. Even after a major revision addressing all points raised below, I recommend submitting it to a different journal (for example, the mathematical biology section of the MDPI Mathematics rather than MDPI Biology) since I believe the model cannot capture biological reality.

Major concerns:

  1. The mathematical analysis described in Methods (Equations 1 to 18) is a textbook material of a drift-diffusion process (One can find it in standard textbooks such as "A Handbook of Stochastic Methods" by C W Gardiner; "A Guide to First-Passage Processes" by Sidney Redner). This part is not novel and can be extensively shortened with appropriate citations. It is unclear what novel contributions are brought to the field! Importantly, the authors should highlight the novel parts of the study and discuss them in the Introduction and Discussions. 
  2. Most studies indicate that the cell membrane is a crowded environment where a simple drift-diffusion model cannot describe the diffusion of a protein; instead, the framework of anomalous diffusion should be used. This is a critical flaw in mathematical modeling. In fact, the authors themselves have cited a paper (Dan V. Nicolau Jr. et al., Biophys J, 2007; and the references therein) that advocates the anomalous diffusion process (also see S Ramadurai et al., JACS, 2009).
  3. As stated in point 2, the Langevin equation (Eq. 17) is inadequate to describe anomalous diffusive behavior. For a Langevin dynamics framework describing a Levy walk (a kind of anomalous diffusion), please see Hans C. Fogedby, Phys Rev E, 1994; and Xudong Wang et al., New J Phys, 2019. Such dynamics could be more appropriate in this case. Authors could redo the numerical calculations using such a framework.

Minor concerns (but still important):

  1. Figure 2: There are no labels for the force values (F), and the curves are not readable! All panels show similar curves! To highlight exponential increase (Eq. 19), the authors should plot up to large times (maybe in semi-log plots).
  2. The paper needs to be better written. For instance, the very first sentence in the Introduction reads: "The main aim of this paper is to offer a mathematical description of cell-to-cell communication and to supplement the biological understanding of cell-to-cell communication by introducing a model that describes cell-to-cell communication." This sentence contains the word "cell-to-cell communication" multiple times that is redundant! There are many such examples that authors should catch and correct them.
  3. The abstract should be more short and succinct.
  4. As stated before, the Introduction or Discussion should highlight the study's novelty and how the modeling differs from previous theoretical studies.

Please see the point 2 in 'minor concerns' (in the above review report). 

Author Response

(The authors gave the same response as above.)

Reviewer 3 Report

The authors describe in detail the randomness exhibited by proteins during cell-to-cell communication. The authors showed solid theory and rigorous derivation, and authors draw valid conclusions. My advice is to accept it.

Author Response

The authors would like to thank the reviewer their comments. 

Many thanks,

Reviewer 4 Report

The manuscript describes the random motion of protein molecules in cell membrane. In this manuscript, a simple one-dimensional lattice random model is used to obtain the position equations of protein molecules in random motion, and the properties under different external forces and drag coefficients are discussed. The overall frame of the manuscript is reasonable, but the manuscript itself is not very innovative. However, if the author can make the following modifications, the article can be accepted. Suggestions to improve the paper:

1.   What the manuscript really discusses is the 17th and 30th formulas. The previous 16 formulas have not been applied. I don't know why the authors listed these 16 formulas in detail. The 17th formula, in particular, doesn't have much of a logical relationship with the previous formula.

2.   The 17th formula should specify where it comes from, especially the parameters it contains, and its physical meaning should be clearly stated. K is k?

3.   How does the model take into account the properties of cell membranes and proteins? In other words, would it be the same if protein molecules were different particles and cell membranes were different interfaces? The author should make some remarks about their particularity.

4.    

5.   The Figures are not very normalized and the units in the axes are not indicated.

6.   The Abstract section is too long and the conclusion section is too short. The author is advised to make appropriate adjustments.

Extensive editing of English language required

Author Response

(The authors gave the same response as above.)

Round 2

Reviewer 2 Report

Some parts of the manuscript have been improved following the suggestions. However, the main concern still needs to be addressed. The author should justify in the Discussion why the framework of anomalous diffusion was not used. 

None.

Reviewer 4 Report

The author has responded to all my concerns, please publish as is